# Predictive Biomarkers for Checkpoint Inhibitor Immune-Related Adverse Events

**DOI:** 10.3390/cancers15051629

**Published:** 2023-03-06

**Authors:** Iñigo Les, Mireia Martínez, Inés Pérez-Francisco, María Cabero, Lucía Teijeira, Virginia Arrazubi, Nuria Torrego, Ana Campillo-Calatayud, Iñaki Elejalde, Grazyna Kochan, David Escors

**Affiliations:** 1Internal Medicine Department, Navarre University Hospital, 31008 Pamplona, Spain; 2Autoimmune Diseases Unit, Internal Medicine Department, Navarre University Hospital, 31008 Pamplona, Spain; 3Inflammatory and Immune-Mediated Diseases Group, Instituto de Investigación Sanitaria de Navarra (IdISNA), Navarrabiomed-Public University of Navarre, 31008 Pamplona, Spain; 4Osakidetza Basque Health Service, Department of Medical Oncology, Araba University Hospital, 01009 Vitoria-Gasteiz, Spain; 5Lung Cancer Research Group, Bioaraba Health Research Institute, 01006 Vitoria-Gasteiz, Spain; 6Breast Cancer Research Group, Bioaraba Health Research Institute, 01006 Vitoria-Gasteiz, Spain; 7Clinical Trials Platform, Bioaraba Health Research Institute, 01006 Vitoria-Gasteiz, Spain; 8Medical Oncology Department, Navarre University Hospital, 31008 Pamplona, Spain; 9Oncoimmunology Group, Instituto de Investigación Sanitaria de Navarra (IdISNA), Navarrabiomed-Public University of Navarre, 31008 Pamplona, Spain

**Keywords:** immune-related adverse events, immune-checkpoint inhibitors, biomarkers, prediction, diagnosis

## Abstract

**Simple Summary:**

Immune-checkpoint inhibitors (ICIs) are increasingly used in the treatment of cancer, but they cause immune-related adverse events (irAEs) in around 40% of patients treated. Identifying biomarkers predictive of irAEs has become a priority for the optimal management of patients on ICIs. Herein, we review the state of the art regarding the most relevant biomarkers for predicting irAEs, distinguishing between biomarkers already clinically available and those under investigation. Although none of these biomarkers has been validated in prospective studies, there is growing evidence supporting their use for irAE prediction and clinical characterization, which depend on cancer type, ICI agent and organ affected by the toxicity. A better understanding of the pathogenic mechanisms underlying irAEs and the combination of different emerging biomarkers would allow us to improve the risk-benefit balance for patients who are candidates for ICI therapy.

**Abstract:**

Immune-checkpoint inhibitors (ICIs) are antagonists of inhibitory receptors in the immune system, such as the cytotoxic T-lymphocyte-associated antigen-4, the programmed cell death protein-1 and its ligand PD-L1, and they are increasingly used in cancer treatment. By blocking certain suppressive pathways, ICIs promote T-cell activation and antitumor activity but may induce so-called immune-related adverse events (irAEs), which mimic traditional autoimmune disorders. With the approval of more ICIs, irAE prediction has become a key factor in improving patient survival and quality of life. Several biomarkers have been described as potential irAE predictors, some of them are already available for clinical use and others are under development; examples include circulating blood cell counts and ratios, T-cell expansion and diversification, cytokines, autoantibodies and autoantigens, serum and other biological fluid proteins, human leucocyte antigen genotypes, genetic variations and gene profiles, microRNAs, and the gastrointestinal microbiome. Nevertheless, it is difficult to generalize the application of irAE biomarkers based on the current evidence because most studies have been retrospective, time-limited and restricted to a specific type of cancer, irAE or ICI. Long-term prospective cohorts and real-life studies are needed to assess the predictive capacity of different potential irAE biomarkers, regardless of the ICI type, organ involved or cancer site.

## 1. Introduction

In recent years, treatment with immune checkpoint inhibitors (ICIs) has led to a paradigm shift in the treatment of various types of cancer [1,2]. The mechanism of action of ICIs consists of blocking certain inhibitory receptors in the immune system, such as the cytotoxic T-lymphocyte-associated antigen 4 (CTLA-4), programmed death cell protein 1 (PD-1), and PD ligand 1 (PD-L1). By blocking these inhibitory pathways, ICIs induce an enhanced T-cell-mediated response aimed at eliminating tumor cells. As a result of this immune overactivation, ICIs may also trigger a wide range of toxic effects known as immune-related adverse events (irAEs) [3], which mimic traditional autoimmune disorders. In practical terms, an irAE can be defined as any symptom, sign, syndrome, or disease caused or exacerbated by an immune-activating mechanism during the administration of an ICI once other causes such as infectious diseases or tumor progression have been ruled out [4]. The burden of irAEs is high because they are common and, not infrequently, severe complications impact the quality of life and prognosis of patients receiving ICIs [5]. Furthermore, it remains unclear how best to manage irAEs without interfering with ICI-related antitumor response and long-term patient survival [6]. Indeed, patients who develop irAEs have a better cancer-related prognosis [7,8,9]. Therefore, it is of great interest to assess the individual risk of toxicity in advance, allowing earlier management of irAEs, which would help maintain ICIs in these patients susceptible to immune-mediated complications but who, paradoxically, benefit more from therapy.

With the progressive expansion of ICI use in the oncology field, there is an increasing need for reliable and validated biomarkers able to predict irAEs [10]. In accordance with FDA guidelines, a biomarker is “a defined characteristic that is measured as an indicator of normal biological processes, pathogenic processes or responses to an exposure or intervention” [11]. In line with this, a predictive biomarker can be defined as a factor that is “used to identify individuals who are more likely than similar individuals without the biomarker to experience a favorable or unfavorable effect from exposure to a medical product or an environmental agent” [12]. Recent years have seen a proliferation of studies on predictive biomarkers for irAEs. Nonetheless, the clinical benefit of reported biomarkers still needs to be confirmed by long-term prospective studies, preferably within randomized clinical trials or real-life studies. 

To date, most research on irAE biomarkers has had similar shortcomings: a short follow-up time, retrospective design, and a restricted focus on specific types of irAE, ICI or cancer. That is, there is a lack of long-term, multicenter, and prospective studies encompassing pan-tumor cohorts of patients treated with different ICI agents. Moreover, a cross-sectional use of generic predictors aimed at different tumors, irAEs and ICIs is not feasible because each organ or system damaged by immune toxicity is related to a specific biomarker. For instance, it is well known that the risk of developing nivolumab-induced destructive thyroiditis is higher in patients with antithyroid antibodies pre-treatment [13]. Strikingly, some studies have suggested that certain specific autoantibodies, such as antithyroid antibodies, may herald the risk of irAEs at other anatomic sites [14]. Overall, current knowledge gaps on the pathogenesis of irAEs prevent us from estimating individual patients’ risk of ICI-mediated toxicity and motivate us to search for more effective predictive biomarkers. 

Among the wide range of potentially useful biomarkers, we can distinguish between biomarkers that, though not validated, are available for routine clinical use and investigational biomarkers [15]. The aim of this paper is to review the literature on predictive biomarkers for irAEs from a practical approach, differentiating between biomarkers already available for use in daily practice and those still at the research stage.

## 2. Material and Methods

The search strategy is detailed in Appendix A.

## 3. Results

First, it should be noted that no biomarkers have yet been validated as an irAE predictor in asymptomatic patients treated with ICIs [16]. Taking a pragmatic, clinically based approach, we have classified potentially predictive irAE biomarkers as those currently available for clinical use and those still under investigation (Figure 1). Within this classification, clinically available biomarkers would be easily accessible to attending physicians if validated for this purpose, while those still under investigation would require implementation in clinical practice in addition to validation.

### 3.1. Biomarkers Available for Clinical Use

#### 3.1.1. Autoantibodies

The potential use of autoantibodies as predictive biomarkers of irAEs has become an expanding field of research [17,18]. Currently, guidelines do not recommend testing every patient for autoantibodies before ICI initiation as this indicator has not been validated for irAE screening [19]. The association between autoantibodies and irAEs is, however, well documented in the case of organ-specific irAEs and the autoantibodies related to such events [20,21,22]. For instance, the risk of suffering ICI-induced thyroiditis is higher in patients with pre-existing antithyroid antibodies [13,23,24]. That is, on the one hand, some organ-specific autoantibodies are only useful for organ-specific irAEs, although not all reported irAEs have been paired with a specific autoantibody. On the other, generic and routinely available autoantibodies, such as antinuclear antibodies (ANA) or rheumatoid factor, may be useful in screening for any type of irAE, regardless of the tissue involved [25]. Furthermore, even certain organ-specific autoantibodies such as antithyroid antibodies, traditionally linked to the prognosis of immunogenic tumors [26], could also be useful in predicting irAEs at any site, indicating a marked overlap between generic and specific autoantibodies [14,25]. Nevertheless, the clinical heterogeneity and complex and diverse pathogenesis, as well as the generally low rates of autoantibody seropositivity associated with these events, mean that current autoantibody panels are not applicable to all patients developing ICI-related toxicity.

In recent years, research on autoantibodies as irAE indicators has moved from retrospective towards prospective methods (Table 1) [14,25,27,28,29,30,31,32,33,34,35,36,37,38,39]. While testing positive for autoantibodies at baseline was considered a risk factor for irAE development in preliminary studies, the most recent reports have provided greater insight into changes in antibody levels over time. For example, a retrospective study by Toi et al. suggested that patients with pre-treatment ANAs, rheumatoid factor or antithyroid antibodies were at increased risk of developing irAEs [27]. De Moel et al. showed an association between seroconversion of any autoantibody included in a battery of 23 autoantibodies and irAEs during follow-up, especially when focusing on specific irAEs related to the battery under study; however, the presence of autoantibodies before ICI initiation was not associated with irAEs [30]. The value of autoantibody seroconversion was also highlighted by Giannicola et al., who found a higher risk of irAEs in patients who became positive for ANAs, anti-extractable nuclear antigens antibodies or anti-smooth muscle antibodies after starting nivolumab administration [31].

Confirming the significance of dynamic changes in autoantibody titer, a sub-study from a phase II clinical trial identified a low autoantibody titer at baseline and greater fold change in autoantibody titer after ICI initiation as independent risk factors for irAEs [39], in contrast with the pre-formed autoantibody theory. Moreover, Alsewaran et al. demonstrated that pre-treatment ANA positivity was not associated with irAE development. On the contrary, patients experiencing seroconversion to ANA positivity after ICI initiation developed more severe irAEs than patients who remained ANA-negative and patients who were ANA-positive before ICI initiation. This humoral response in the form of ANA seroconversion could be related to early B-cell changes induced by ICIs, namely a decline in circulating B cells and an increase in CD21 B cells and plasmablasts, which have been associated with a higher frequency of irAEs [40]. Furthermore, in up to 83% of patients with severe irAEs, modifications in ANA patterns preceded irAE onset [38]. In addition to ANA positivity, ANA patterns determined by immunofluorescence may be useful for discriminating between primary autoimmune diseases and irAEs. Although no studies designed to compare autoantibody profiles have yet been reported, patients who develop irAEs may be less likely to express disease-specific ANA patterns than patients with the corresponding classical autoimmune disease. In contrast, a nuclear speckled pattern may be more indicative of immune-related toxicity [38].

#### 3.1.2. Blood Cell Counts and Ratios

The use of blood cell counts for the early detection of irAEs may be of great interest to clinicians due to their wide availability, low cost, and easy interpretation. Although not completely consistent, there is supportive evidence suggesting that baseline absolute neutrophil, lymphocyte, monocyte, eosinophil and basophil counts, platelet counts, and increases in white blood cell, lymphocyte and eosinophil counts during follow-up are associated with a higher risk of irAEs (Table 2) [41,42,43,44,45,46,47,48,49,50,51,52,53,54,55,56,57,58,59,60]. In addition, several blood cell ratios, the most common being the neutrophil-to-lymphocyte ratio (NLR) and derived NLR (calculated as absolute neutrophil count/[white blood cell count–absolute neutrophil count]), could help to predict irAEs before and after ICI initiation. In a systematic review and meta-analysis including 6696 patients on ICIs from 25 studies, a high NLR was identified as an independent risk factor for developing irAEs [51]. In a prospective study including 1187 patients, an elevated NLR at the beginning of ICI therapy was predictive of very severe irAEs (grades 4 and 5) [42].

Similarly, it has been reported that peripheral CD8 T-cell expansion and diversification after ipilimumab initiation, as surrogate markers of autoreactivity against tissue self-antigens at the systemic level, allows us to predict irAE onset with very high sensitivity [61]. This diversification occurs early in follow-up, within the first two weeks after ICI administration [62]. Furthermore, patients experiencing ipilimumab-induced colitis showed higher absolute counts of peripheral CD4+ T cells and lower percentages of regulatory T cells at baseline [63]. Likewise, it has been suggested that changes in the percentage of peripheral CD4+ CD25+ Foxp3+ regulatory T cells, a cell subset in charge of maintaining immune tolerance in the tumor microenvironment, may be predictive of irAEs [64]. Recently, elevated levels of circulating low-density neutrophils, a myeloid subpopulation with immunosuppressive properties, have been associated with a poor response to pembrolizumab mediated by T-cell cytotoxicity down-regulation in patients with non-small cell lung cancer [65]. As in tumor response, different subpopulations may play a role in the pathogenesis of ICI toxicity.

Despite these promising findings, most published studies on blood cell counts and ratios have been retrospective, time-limited (usually considering only baseline data or short follow-up periods) and constrained to either a specific type of cancer, irAE or ICI agent (Table 2). Notably, to date, few prospective studies have assessed the clinical value of blood cell count fluctuations for predicting irAEs in the long term.

#### 3.1.3. Serum and Other Biological Fluid Proteins

Baseline levels of thyroid-stimulating hormone in serum have been shown to predict immune-related thyroiditis before ICI initiation [66,67,68]. Likewise, thyroid stimulating hormone is the most efficient biomarker for monitoring thyroid dysfunction in patients on ICI therapy [6]. Similarly, serial measurements of serum brain natriuretic peptide and troponin, together with new-onset electrocardiographic abnormalities, help anticipate cardiovascular irAEs [69]. Fecal lactoferrin and calprotectin are commonly used as screening tools for ICI-induced colitis [70], calprotectin being a good non-invasive indicator for assessing treatment response and avoiding repetitive endoscopic procedures [71].

Among generic biomarkers, raised C-reactive protein levels correlate well with the risk of developing irAEs [51], in parallel with serum interleukin-6 (IL-6) levels [72]. With different cut-offs, high serum albumin levels have also been associated with irAEs [48,73]. In addition, elevated blood lactate dehydrogenase levels predispose patients to irAEs [74], especially high-severity events (grade ≥ 3) [48]. In contrast, a decrease in serum leptin levels at four weeks from ICI initiation was more common in patients who experienced irAEs than in irAE-free patients [75]. Levels of these generic proteins, which are acute-phase reactants or tumor burden-related markers, are relatively easy to interpret in patients with an indication for adjuvant treatment. In contrast, it may be less appropriate to use them as irAE biomarkers in patients with metastatic disease since cancer, especially in progressive phases, may alter protein levels. Interestingly, a reduction over time in the level of certain serum tumor markers, such as the melanoma-inhibitory activity protein, could help discriminate between toxicity and progression in patients with metastatic melanoma [72,76].

#### 3.1.4. Cytokine Profiles and Dynamics

In recent years, the interest in cytokines to predict irAE susceptibility has grown steadily [77]. Certain cytokine profiles at baseline and dynamic fluctuations in cytokine levels over time have been associated with a higher risk of developing irAEs and better treatment outcomes [78]. Moreover, the uncoupled effect achieved by some anti-cytokine drugs in preclinical studies, consisting of decreased ICI-induced toxicity without sacrificing antitumor activity, makes it a priority to improve our understanding of the pathogenic role of cytokines [79,80]. Unfortunately, not all cytokines are currently assessed in clinical practice for diagnostic purposes. In addition, the cytokines involved in a particular irAE can differ from the corresponding autoimmune manifestation and with the ICI agent administered [81].

Tumor necrosis factor-α (TNF-α) is one of the most studied biomarkers in the field of irAE research [82]. Low baseline TNF-α levels may predispose patients to better antitumor immunity [83], while it is unknown whether fluctuations in TNF-α levels over time can anticipate irAE onset. In any case, various TNF-α blockers, such as infliximab, etanercept, adalimumab and certolizumab, have been used as rescue therapy for steroid-refractory cases of ICI-induced colitis, arthritis and pneumonitis [84,85,86,87]. Two concerns arising from TNF-α antagonism are the attenuation of antitumor immunity and promotion of tumorigenesis by anti-TNF-α drugs [88], which may depend on the dose and duration of treatment. Indeed, it is accepted that short courses of TNF-α inhibitors given at regular doses are safe for patients undergoing ICI therapy [89]. Moreover, preclinical data suggesting an antitumor benefit in mice combining ICIs and TNF-α inhibitors warrant the undertaking of clinical trials assessing this hypothesis (NCT03293784) [90,91].

Considered a “usual suspect”, IL-6 is a proinflammatory cytokine that is potentially involved in the pathogenesis of several immune-mediated disorders [92]. Low baseline levels of IL-6 were strongly associated with irAEs [93,94]. Combined with C-reactive protein, IL-6 has also been proposed as an early biomarker for irAE detection during follow-up [72,83]. Even in patients with elevated C-reactive protein, regardless of serum IL-6 levels, IL-6 blockade with tocilizumab has been tested as a therapeutic and pre-emptive drug for irAEs [95]. Recently, an uncoupled effect on induced toxicity and antitumor immunity exerted by immunotherapy has been achieved by the blockade of IL-6 in a murine model [96], which has warranted the launching of a phase II clinical trial to assess the efficacy of tocilizumab in patients receiving ICIs (NCT04940299). The favorable safety profile of tocilizumab and other anti-IL-6 agents, widely available in the clinical setting due to the SARS-CoV-2 pandemic, makes IL-6 a promising therapeutic target [97].

Interleukin-17 (IL-17) is another pro-inflammatory cytokine involved in the pathogenesis of inflammatory bowel disease, psoriasis, psoriatic arthritis, other types of spondyloarthritis, and even interstitial lung disease [98,99]. Unlike TNF-α and IL-6, high serum levels of IL-17 at baseline have been associated with severe colitis in patients on ipilimumab [100]. An increase in serum IL-17 levels was demonstrated following CTLA-4 blockade with ipilimumab in patients developing colitis, consistent with CTLA-4 inhibiting the production of IL-17 by type 17 T helper (Th17) cells [101,102]. These findings support the central role of IL-17 in the pathogenesis of irAEs involving Th17 cell-enriched tissues, such as colitis, psoriasiform dermatitis, pneumonitis and neuroendocrine toxicity [103,104,105]. Correspondingly, IL-17 antagonists have shown a clinical benefit in IL-17-dependent irAEs, opening the door to targeted anti-cytokine therapies [106].

The case of IL-1 is revealing and may represent a new therapeutic target. There is evidence suggesting that baseline elevated levels of IL-1β are related to thyroid dysfunction [107]. In a retrospective series, IL-1α was significantly elevated in patients who developed ICI-induced myositis [108]. Notably, patients treated with a combination of anti-CTLA-4 and anti-PD-1 drugs who developed ICI-induced colitis overexpressed mucosal IL-1β (as well as IL-17), but not TNF-α, with a higher abundance of *Bacteroides intestinalis* [109]. Moreover, fecal microbiota transplant of large numbers of *Bacteroides intestinalis* bacteria to mice induced overexpression of IL-1β after ICI administration [5]. These findings on the role of IL-1 regardless of TNF-α activity in immune-mediated colitis could be related to the subgroup of patients with inflammatory bowel disease who are refractory to standard therapy with TNF-α blockers [110]. In addition, IL-1β has been identified as an independent risk factor for irAEs in patients on PD-(L)1 inhibitors [111].

The pro-inflammatory interleukins IL-12 and IL-23 belong to the IL-12 family and are characterized by sharing a p40 subunit. Blocking both IL-12 and IL-23 with ustekinumab has been shown to be effective in ICI-induced refractory colitis [112]. On the other hand, interfering with the IL-12-dependent pathway may alter the antitumor effect associated with ICI therapy [113,114]. The use of guselkumab, a specific anti-IL-23 agent, has been proposed as a way of inhibiting both the pro-tumor and pro-inflammatory effects attributed to IL-23 without affecting the IL-12 pathway, although this hypothesis needs further testing [115].

Regarding IL-10, an anti-inflammatory interleukin with homeostatic properties [116], a retrospective study revealed that high baseline IL-10 levels and increases in these levels after the first cycle of ICI were the only independent factors predicting irAEs among a broad battery of cytokines [117]. Further insight is required into the mechanisms by which IL-10 may promote immune tolerance and their relationship with toxicity modulation [118].

Other potential cytokine-related biomarkers are being studied, such as the serum soluble IL-2 receptor, a biomarker of hyper-inflammatory status available in daily clinical practice [119]. In addition, low baseline values and decreases in interferon-γ release, commonly used for the detection of *Mycobacterium tuberculosis* (latent) infection, have been associated with ICI-induced pneumonitis [120].

### 3.2. Biomarkers under Investigation

#### 3.2.1. Other Cytokines and Serum Proteins under Development

The chemokine ligand 15 (CXCL15) and the soluble protein cluster of differentiation 163 (sCD163), as surrogate indicators of Th17 cell and tumor-associated macrophage activation, respectively, have been proposed as biomarkers for irAE prediction [121]. Lower levels of CXCL9, CXCL10, CXCL11 and CXCL19 at baseline and greater increases in CXCL9 and CXCL10 levels have been reported in patients who experienced irAEs [122]. Other biomarkers under investigation are angiopoietin-1 (Ang-1) and CD40 ligand, whose baseline high levels have been related to dermatitis [78]. Early decreases in granulocyte colony-stimulating factor have been associated with several irAEs, while lower baseline levels of this growth factor may predispose patients to colitis [78,107]. High baseline growth-regulated oncogene-1 and granulocyte macrophage colony-stimulating factor levels have been associated with generic irAEs and specifically with thyroid dysfunction and dermatitis, respectively [75,107]. Other organ-specific irAEs were associated with stem cell factor (colitis), leukemia inhibitory factor and placental growth factor (both with myositis), and B and T-lymphocyte attenuator (dermatitis) [108].

In addition, the pathogenic role and predictive value of other cytokines such as IL-2, IL-4, IL-5, IL-15, IL-27, IL-35, and interferon-α remain to be clarified [107,108,123,124].

Overall, the incorporation of several cytokines, both those available routinely and those under development, into already-designed toxicity risk scores such as the CYTOX score might provide a useful tool for irAE prediction [125]. We are currently witnessing a surge in precision medicine approaches based on personalized cytokine profiles depending on individual, pharmacologic and tissue-related factors, without undermining the antitumor response. Such molecular-focused strategies yield therapies focused on specific cytokine signatures rather than the targeted organ [126].

#### 3.2.2. Genetic Variations and Gene Expression Profiling

Monogenic mutations leading to autoimmune diseases have been identified for years [127]. For instance, we know about the existence of cases of systemic lupus erythematosus caused by monogenic mutations in the C1qA, B, C, C1R, DNASE1, DNASE1L3, and ACP5 genes [128], among many other predisposing genetic variations [129]. Furthermore, certain germline CTLA4 and PDCD1 (encoding for PD-1 protein) gene polymorphisms have been associated with both the development of autoimmune diseases and susceptibility to ICI-induced endocrine irAEs [130] (Table 3) [131,132,133,134,135,136,137,138,139,140,141,142,143,144]. In the same vein, two different PDCD1 gene single-nucleotide polymorphisms (SNPs), namely rs2227981 and rs10204525, have been identified as protective and susceptibility biomarkers for irAEs, respectively [135,136]; these apparently paradoxical findings are explained by the polymorphism in question, which determines the level of PD-1 expression (low in the case of rs2227981 and high in the case of rs10204525). In a comprehensive study, Abdel-Wahab et al. described as many as 30 SNPs related to irAEs, of which twelve led to a higher irAE risk, eighteen to a lower irAE risk, and nine involved genes associated with autoimmune or inflammatory diseases (GABRP, DSC2, BAZ2B, SEMA5A, ANKRD42, PACRG, FAR2, ROBO1 and GLIS3) [132]. These SNPs add to others previously described as isolated risk factors for irAEs or combined predictors of irAEs and autoimmune diseases [133,145].

In addition, genetic alterations other than SNPs such as small sequence variations and copy number variations (namely, duplications and depletions) have been detected in 16 genes (AIRE, TERT, SH2B3, LRRK2, IKZF1, SMAD3, JAK2, PRDM1, CTLA4, TSHR, FAN1, SLCO1B1, PDCD1, IL1RN, CD274, and UNG) and linked to irAEs affecting different organs and systems [131]. Moreover, patients showing modifications of CEBPA, FGFR4, MET or KMT2B genes detected in circulating tumor DNA before ICI initiation are at higher risk of experiencing irAEs [134].

Another gene-related biomarker for ICI-mediated toxicity is the expression of specific gene signatures. For example, Friedlander et al. proposed a 16-gene signature (involving CARD12, CCL3, CCR3, CXCL1, F5, FAM210B, GADD45A, IL18bp, IL2RA, IL5, IL8, MMP9, PTGS2, SOCS3, TLR9 and UBE2c genes) to discriminate between low- and high-grade tremelimumab-induced diarrhea [139]. Clinically relevant pathways, such as those of the inflammasome in ICI-induced myocarditis or the neutrophil activation cascade in gastrointestinal irAEs, have been identified through the overexpression of type 5 and 6 guanylate binding proteins and CD177 and CEACAM1 genes, respectively [140,141]. More specifically, IFI27 gene expression, related to the interferon-α pathway, has allowed ICI-associated T cell-mediated rejection to be distinguished from ICI-associated acute interstitial nephritis in kidney transplant patients [142].

With the assistance of pharmacovigilance, other over-represented genes in patients with irAEs have been identified through integrated bioinformatic analysis [143], molecular multi-omics data [144], and transcriptomic information of messenger RNA and alternative splicing features [146,147] (Table 3). Again, further studies are needed to validate these promising results in clinical practice.

#### 3.2.3. Human Leucocyte Antigen Genotyping

Among the genes most influential in irAEs are those in the major histocompatibility complex, also known as the human leucocyte antigen (HLA) system, which is a group of genes encoding for surface glycoproteins involved in antigen presentation that has been widely related to susceptibility to immune-mediated diseases and cancer [148,149]. Although clinically available, the use of HLA genotyping for diagnostic purposes is constrained to specific disorders such as celiac disease or axial spondyloarthritis [150,151]. Despite other confirmed genotypic-phenotypic associations, testing is not recommended for entities with more reliable diagnostic methods, because HLA variants indicate a genetic susceptibility rather than a diagnosis of certainty.

The potential association between certain HLA genotypes and polymorphisms and the risk of immune-related toxicity has been mainly assessed in the context of endocrinologic irAEs, such as ICI-induced diabetes (overall, the irAE most studied from the point of view of the HLA system) [152], thyroid dysfunction and hypophysitis [13,153]. For instance, the development of ICI-induced type 1 diabetes mellitus [154,155], thyroiditis [156] and even autoimmune polyglandular syndrome type 2 [157] was observed in patients with HLA-DR4 more often than in patients with another HLA haplotype. Notably, Delivanis et al. demonstrated that ICI therapy can increase HLA-DR surface expression in activated monocytes leading to pembrolizumab-induced thyroiditis [158].

Other relevant associations of HLA alleles or proteins with irAEs previously reported are those of HLA-DRB1*04:05 with inflammatory arthritis [159], HLA-B27*05 with autoimmune encephalitis [160], HLA-Cw12 with hypophysitis [153,161], HLA-DQB1*03:01 with colitis [162], HLA-DRB3*01:01 with thrombocytopenia [163], HLA-A03 with pneumonitis [164], and HLA DRB*04:01, HLA-DRB1*15:01 and HLA-DQB1*06:02 with hepatitis [165] (Table 3) [74,131,152,153,154,155,156,157,159,160,161,162,164,165,166,167,168,169]. By contrast, some studies have reached negative results regarding the connection between irAEs and the HLA system [170,171].

Due to its growing availability, HLA genotyping could be considered an irAE biomarker on the boundary between clinical and investigational.

#### 3.2.4. Micro-RNAs

A micro-RNA (miR) is a non-coding molecule of single-stranded RNA containing between 20 and 25 nucleotides, which can regulate the post-transcriptional expression of genes by blocking translation of targeted messenger RNA through a process known as ribo-interference [172]. Like CTLA-4 and PD-1, certain miRs, such as miR-146a, promote down-regulation of both innate and adaptive immune responses and may impact ICI-related survival [173], in part by counteracting cell escape mechanisms in the tumor microenvironment [174]. Indeed, the first phase 1 clinical trial which evaluated a liposomal mimic of miR-34a was halted early due to severe irAEs in participating patients [175].

Among the most relevant miRs (Table 3) [176,177,178], miR-146a is a miR family whose modified expression has been involved in the pathogenesis of several autoimmune diseases, including rheumatoid arthritis, psoriasis, and laboratory-induced colitis [179,180,181]. Moreover, specific miR-146a SNPs, such as rs2910164, predispose patients on ICIs to a higher risk of developing severe irAEs and reduced progression-free survival [177]. By contrast, exogenous administration of a miR-146a mimic may mitigate irAE intensity assessed by histopathologic criteria in mice [176].

**Table 3 cancers-15-01629-t003:** Summary of studies on reported associations between immune-related adverse events and HLA system antigens, genetic variants and signatures and micro-RNAs.

Genetic Variants and Gene Expression Profiles					
Reference	Study Design (No. Patients)	Type of Tumor	Type of irAE	Associations	
Wölffer M. [131]	Prospective (*n* = 95)	Melanoma	All types	VARs on SMAD3 gene	Pancreatitis
				CNVs on IL1RN and deletions on PRDM1 genes	Higher risk of irAEs
				Duplications on CD274 and CNVs on SLCO1B1 genes	Hepatitis
				CNVs on PRDM1 and CD274 genes	Encephalitis
				CNVs on PRDM1, CD274, TSHR and FAN1 genes	Myositis
Abdel-Wahab N. [132]	Retrospective	Melanoma	All types	Several SNPs on GABRP, DSC2, BAZ2B, SEMA5A, OSBPL6, AGPS and LOC102724355, and near CFAP65 and LOC100129175 genes	Higher risk of irAEs
	(*n* = 89)			Several SNPs on LOC105377125, RGMA, ANKRD42, PACRG, FAR2, LOC105374140, ROBO1, GLIS3, PVT1, PACRG and PREX2 genes	Lower risk of irAEs
Refae S. [133]	Retrospective (*n* = 94)	Pan-tumor	All types	Several SNPs on UNG, IFNW1, PD-L1, IFBL4 and CTLA-4 genes	Higher risk of irAEs
Jin Y. [134]	Retrospective (*n* = 46)	Gastric cancer	All types	Alterations in CEBPA, FGFR4, MET or KMT2B genes ^#^	Higher risk of irAEs
Bins S. [135]	Retrospective (*n* = 322)	NSCLC	All types	Homozygous 804C > T (rs2227981) SNP on PDCD1 gene	Lower risk of any grade of irAEs
Kobayashi M. [136]	Retrospective (*n* = 106)	Renal cell cancer	All types	PD-1.6 SNP (G allele) on PDCD1 gene (rs10204525)	Higher risk of severe and multiple irAEs
Khan Z. [137]	Retrospective (*n* = 479)	Bladder cancer	Skin irAEs	Genetic variants related to vitiligo and psoriasis, assessed by a polygenic risk score	Higher risk of irAEs and better survival
Khan Z. [138]	Retrospective (*n* = 6075)	Pan-tumor	Thyroid dysfunction	Genetic variants related to autoimmune hypothyroidism, assessed by a polygenic risk score	Higher risk of irAEs and better survival
Friedlander P. [139]	Prospective (*n* = 150)	Melanoma	Diarrhea/colitis	Gene signature composed of 16 inflammation-related genes (CARD12, CCL3, CCR3, CXCL1, F5, FAM210B, GADD45A, IL18bp, IL2RA, IL5, IL8, MMP9, PTGS2, SOCS3, TLR9, UBE2C)	Differentiation between grade 0–1 and grade 2–4 diarrhea
Sahabi V. [140]	Prospective (*n* = 162)	Melanoma	Gastrointestinal irAEs	Increase in expression of CD177, CEACAM1 and immunoglobulin-related genes (IGHA1, IGHA2, IGHG1, and IGHV4–31)	Higher risk of gastrointestinal irAEs
Finke D. [141]	Retrospective (*n* = 19)	All types	Myocarditis	Upregulation of 3784 genes with overexpression of interferon-γ and inflammasome-regulating proteins (GBP5 and 6)	Higher risk of myocarditis
Adam BA. [142]	Retrospective (*n* = 75) *	All types	AIN	Overexpression of IFI27 gene (related to interferon-α)	Discrimination between AIN and TCMR
Zhang Y. [143]	Preclinical study (*n* not applicable)	Not applicable	Thyroid dysfunction	Overexpression of ALB, MAPK1, SPP1, PPARG and MIF genes	Hypothyroidism
				Overexpression of ALB, FCGR2B, CD44, LCN2, and CD74 genes	Hyperthyroidism
Jing Y. [144]	Retrospective (*n* = 18,706)	Pan-tumor (26 types)	General irAEs	Overexpression of LCP1 and ADPGK genes	Higher risk of irAEs
HLA Antigens					
Reference	Study Design (No. Patients)	Type of Tumor	Type of irAE	Associations	
Kobayashi T. [161]	Retrospective (*n* = 62)	All types	Endocrine irAEs	HLA-Cw12, HLA DR-15, HLA-DQ7 and HLA DPw9	ACTH deficiency
				HLA-Cw12 and HLA-DR15	Hypophysitis
				HLA-DRB3*01:01	Thrombocytopenia
Jiang N. [163]	Retrospective (*n* = 530)	Pan-tumor	All types	HLA-DPB1*04:02	Hypokalemia, hyponatremia, leukopenia and anemia
				HLA-A*26:01	Hyperbilirubinemia
Capelli LC. [159]	Retrospective (*n* = 26)	Pan-tumor	Articular irAEs	HLA-DRB1*04:05	Inflammatory arthritis
Correale P. [166]	Retrospective (*n* = 256, 29 with pneumonitis)	Pan-tumor	Lung irAEs	HLA-B*35 and HLA-DRB1*11	Pneumonitis
Wölffer M. [131]	Prospective (*n* = 95)	Melanoma	All types	HLA class I homozygosity	Hepatitis
Stamatouli AM. [154]	Retrospective (*n* = 27)	Pan-tumor	Endocrine irAEs	HLA-DR4	T1DM
Lo Preiato V. [155]	Retrospective ** (*n* =200)	Pan-tumor	All types	HLA-DR4	T1DM
Inaba H. [167]	Retrospective (*n* = 25)	Pan-tumor	All types	HLA-B*46:01, HLA-C*14:02, HLA-DPA1*0103 and HLA-DPB1*02:01	Higher risk of thyroid dysfunction
				HLA-DPB1*05:01	Lower risk of thyroid dysfunction
Inaba H. [168]	Retrospective (*n* = 871, 7 with T1DM)	Pan-tumor	T1DM	HLA-DPA1*02:02, HLA-DPB1*05:01 and HLA-DRB1*04:05	T1DM
Shi Y. [157]	Retrospective ** (*n* = 26)	Pan-tumor	APST2	HLA-DR4	APST2
Chang H. [160]	Prospective (*n* = 290, 7 with encephalitis)	Breast and bladder cancer	Encephalitis	HLA-B*27:05	Encephalitis
Yano S. [153]	Retrospective (*n* = 11)	Pan-tumor	Pituitary irAEs	HLA-DR15, HLA-B52 and HLA-Cw12	Hypophysitis
Abed A. [164]	Retrospective (*n* = 179)	NSCLC	All types	HLA class I (but not class II) homozygosity	Lower risk of irAEs, especially pneumonitis
				HLA-A03	Higher risk of irAEs
Hasan Ali O. [162]	Prospective (*n* = 102)	NSCLC Melanoma	All types	HLA-DRB1*11:01	Pruritus
				HLA-DQB1*03:01	Colitis
Kotwal A. [156]	Prospective (*n* = 10)	Pan-tumor	Endocrine irAEs	HLA-DR4-DR53 and HLA-DR15	Thyroiditis
Purde MT. [165]	Prospective (*n* = 131, 11 hepatitis)	NSCLC Melanoma	Hepatitis	HLA- DRB1*04:01 and HLA- DRB1*15:01-DQB1*06:02	Hepatitis
Clotman K. [152]	Retrospective ** (*n* = 42)	Pan-tumor	T1DM	HLA-DR3-DQ2, HLA-DRB1*04, HLA-DQB1*03:02, HLA-DR4, HLA-A2 and HLA-DR3DQ3, among others	T1DM
Magis Q. [169]	Retrospective (*n* = 163, 5 with T1DM)	Not available	T1DM	HLA-DRB01*03 or HLA-DRB01*04	T1DM
Micro-RNAs					
Reference	Study Design (Sample Size)	Type of Tumor	Type of irAE	Associations	
Marschner D. [176]	Prospective (*n* = 179)	Pan-tumor	All types	Underexpression of miR-146a by SNP on MIR146A gene (rs2910164)	Higher risk of severe irAEs
Ivanova E. [177]	Prospective (*n* = 86)	ccRCC	All types	Underexpression of miR-146a by SNP on MIR146A gene (rs2910164)	Higher risk of severe irAEs
Xia W. [178]	Mouse model	Not applicable	Myocarditis	Overexpression of miR-34a-5p induced by PD-1 inhibitor-treated macrophages led to cardiac senescence	Higher risk of myocarditis

* The study included 75 kidney biopsies. ** The study consisted of a literature review. ^#^ All of them with a *p* = 0.09. Abbreviations in alphabetical order (except for gene designations): ACTH, adrenocorticotropic hormone; AIN, acute interstitial nephritis; APST2, autoimmune polyglandular syndrome type 2; ccRCC, clear cell renal cell carcinoma; CNV, copy number variation; GBP5, guanylate binding protein 5; GBP6, guanylate binding protein 6; HLA, human leukocyte antigen; irAE, immune-related adverse event; miR, micro-RNA; No, number; NSCLC, non-small cell lung cancer; SNP, single nucleotide polymorphism; TCMR, T cell-mediated rejection; T1DM, type 1 diabetes mellitus; VARs, small variations.

Another micro-RNA called miR-34a-5p, related to cardiac injury by doxorubicin and cardiac senescence [182], was found to be involved in ICI-induced cardiotoxicity in an animal model [178]. Interestingly, miR-34a-5p has been shown to modulate the response of M1 macrophages, CD4+ and CD8+ T cells by downregulation of chemokine signaling, specifically of CXCR3 and its ligands CXCL10 and CXCL11 [183].

#### 3.2.5. Gastrointestinal Microbiome

The term gastrointestinal, or gut, microbiome refers to a complex system of microorganisms, mainly bacteria, that inhabit the intestine establishing a symbiotic relationship with the host and participating in several homeostatic processes that contribute to the host’s health [184]. Already known to be involved in the pathogenesis of several immune-based disorders, especially those related to inflammatory bowel diseases [185,186,187], the gut microbiome has also been shown to modulate both intestinal and non-intestinal irAEs [188]. Gut dysbiosis, characterized by a reduction in microbiome diversity and resulting dominance of certain bacteria in the gut, may increase or decrease the anti-tumoral response and the risk of developing irAEs induced by ICIs [189]. As with other irAEs, patients experiencing ICI-induced colitis appear to have better antitumor responses and cancer-related prognosis [190]. An abundance of *Bacteroidetes* phylum has long been known to be a feature of colitis-resistant patients [191], while a microbiome rich in *Faecalibacterium* and other members of *Firmicutes* is associated with an elevated risk of ICI-related colitis [63]. Recently, a well-designed cohort study has confirmed the impact of specific microbial signatures, enriched with *Lachnospiraceae* spp. and *Streptococcaceae* spp., on certain irAEs [192]. In another prospective cohort study, Chau et al. found that the gut microbiome of patients not developing irAEs was relatively enriched with *Bifidobacterium* and *Desulfovibrio* species [193]. Indeed, over recent years, the collection of intestinal commensal micro-organisms potentially related to irAEs has expanded (Table 4) [63,109,186,188,189,190,191,192,193,194,195,196,197,198,199,200,201,202,203,204].

There are many hypothetical underlying mechanisms, likely interconnected, explaining the contribution of the gut microbiome to immune-related toxicity: dysregulation between pro-inflammatory (i.e., IL-6) and anti-inflammatory (i.e., IL-10) interleukins at local and distant tissues [198]; differentiation, expansion and migration of gut mucosal Th17 cells; inactivation of gut-associated regulatory T cells leading to exacerbation of T-cell effector activities [205]; a role of microbiome metabolites such as short-chain fatty acids [205], polyamine transport system and group B vitamins [181], and microbial fragments such as polysaccharide A [206]. As already mentioned in the “Cytokine” section, overexpression of IL-1β and IL-17, but not of TNF-α, has been demonstrated by two independent groups in samples from patients with ICI-induced colitis [5,109]. All these findings open the door to gut microbiota manipulation using various strategies, such as the administration of antibiotics [86], prebiotics, probiotics, or postbiotics [196,201], or fecal microbiota transplantation [200,207,208,209,210], as well as cytokine-targeted therapies for specific irAEs [211]. Nonetheless, many questions remain unanswered in the complex phenomenon of interaction between the gut microbiome and ICI-related toxicity.

#### 3.2.6. Upcoming Biomarkers for irAE Prediction

Neoantigens are immunogenic peptides derived from tumor-specific genetic alterations and presented to T cells only on the malignant cell surface in the presence of the HLA system [212]. Nowadays, the detection of neoantigens, which are tumor and individual-specific, is being used as a way to design targeted therapies based on T-cell-mediated cytotoxicity with a low incidence of irAEs [213]. One of these neoantigens, namely, napsin A, has already been identified as a lung tumor self-antigen present in both lung malignant cells and ICI-induced inflammatory lung lesions [214]. Furthermore, Tahir et al. conducted a serological analysis of recombinant cDNA expression, a technique designed to identify tumor antigens, resulting in the detection of specific anti-CD74 autoantibodies related to pneumonitis, and anti-GNAL and anti-ITM2B autoantibodies related to hypophysitis [215].

In addition, autoantibody signatures profiled using the HuProt human proteome microarray system, which tests a massive number of proteins, may become a prominent tool for predicting toxicity [216], or even efficacy and toxicity simultaneously [217], in the short-to-medium term. Likewise, by means of a microarray autoantigen panel including 120 autoantibodies, Ghosh et al. showed that patients most likely to experience irAEs had lower baseline autoantibody titers and larger increases in these titers over time [39].

Given the complex nature of irAEs and the difficulty of predicting their onset, an approach based on the combination of different omics disciplines, including radiomics [218], together with real-time big data exploitation is on the horizon [219].

## 4. Conclusions

The field of ICI-related toxicity is evolving rapidly [220]. Nowadays, we are witnessing rapid growth in publications on potentially predictive irAE biomarkers [208]. Despite this boom in research, no biomarkers have yet been validated for clinical use. Except for routine laboratory testing, such as complete blood cell count with differential, glucose, renal and liver function tests and thyroid-stimulating hormone measurements, analysis of other laboratory parameters is not recommended before starting ICI therapy [221,222].

Hence, the first question is, Are any known biomarkers capable of predicting toxicity? and the answer is no, at least for most patients. In certain clinical scenarios, we can use some biomarkers for decision-making. Specifically, in the case of patients with a pre-existing autoimmune disease who would benefit from an ICI, measurement of autoantibodies known to be useful in assessing autoimmune disease activity may be indicated [19]. For instance, a high titer of anti-double-stranded DNA antibodies could represent a risk factor for developing a flare in a patient diagnosed with lupus before starting ICI therapy. Likewise, a progressive increase in anti-double-stranded DNA antibody titers could anticipate a lupus flare once an ICI has been initiated.

The identification of versatile biomarkers is also made more complex by the pathogenic mechanisms involved in irAEs, which are diverse and heterogeneous [223]. Hence, not all biomarkers under study are equally applicable to all patients. A pragmatic approach would be to design risk toxicity scores that are cross-sectionally applicable to different settings and include accessible and understandable biomarkers [125,217]. To our knowledge, there are currently no multi-factor prediction models combining baseline patient characteristics with autoantibody titers, blood cell counts or ratios, and levels of easily measurable cytokines. Besides static predictive models, longitudinal data on biomarker fluctuations, such as blood cell counts, autoantibodies, and cytokines, may provide a more reliable approach to assessing the individual risk of experiencing irAEs. Moreover, incorporation into standard practice of more sophisticated but increasingly widespread diagnostic tools, such as HLA genotyping, and measurement of micro-RNA expression, genetic variation and gene expression, and gut microbiome signatures, will depend on their future validation and availability. In this regard, results from emerging research based on artificial intelligence, big data and machine learning as methods for creating predictive models of toxicity are particularly promising [219,224,225].

From a practical point of view, another question that arises is, Once toxicity appears in a particular patient treated with ICIs, would any of these potential biomarkers be reliable for categorizing this toxicity as immune-mediated? If that were to be the case, biomarkers could be recommended as a support tool for the differential diagnosis of a particular complication. In this clinical situation, certain hormone profiles already represent a valuable test for the identification of immune-related endocrinopathies in routine practice. In addition, some biomarkers such as fecal lactoferrin and calprotectin might suggest an immune-mediated process underlies new-onset diarrhea and reduce the number of endoscopic procedures, especially during the recovery phase from colitis. In a more general sense, the elevation of acute phase reactants, such as C-reactive protein, could also point to an immune-related etiology when faced with a nonspecific clinical picture without a clear diagnosis. Nonetheless, such generic biomarkers of inflammation can be difficult to interpret in the context of advanced cancer or concomitant infection.

In conclusion, a better understanding of the pathogenic mechanisms linked to immune-mediated toxicity and the implementation of long-term, prospective, and real-life studies on irAEs are needed to confirm the validity of numerous biomarkers under investigation and enable their adoption in practice in a wide range of clinical scenarios.

## Figures and Tables

**Figure 1 cancers-15-01629-f001:**
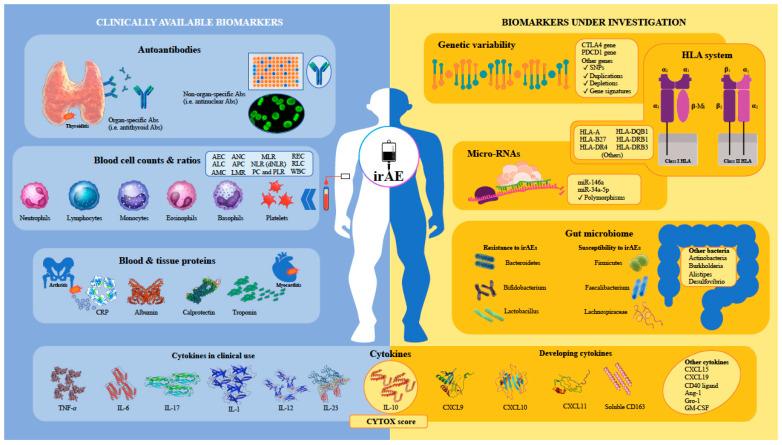
Graphical representation of different families of predictive biomarkers for immune-related adverse events in patients receiving immune-checkpoint inhibitors. Abbreviations in alphabetical order: Abs, antibodies; AEC, absolute eosinophil count; ALC, absolute lymphocyte count; AMC, absolute monocyte count; ANC, absolute neutrophil count; Ang-1, angiopoietin-1; APC, absolute platelet count; CD40, cluster of differentiation 40; CRP, C-reactive protein; CTLA4, cytotoxic T-lymphocyte-associated antigen 4; CXCL, chemokine ligand; dNLR, derived neutrophil-to-lymphocyte ratio; GM-CSF, granulocyte-macrophage colony-stimulating factor; gro-1, growth-regulated oncogene-1; HLA, human leucocyte antigen; IL, interleukin; irAE, immune-related adverse event; LMR, lymphocyte-to-monocyte ratio; miR, micro-RNA; MLR, monocyte-to-lymphocyte ratio; NLR, neutrophil-to-lymphocyte ratio; PC, platelet count; PDCD1, programmed cell death protein 1; PLR, platelet-to-lymphocyte ratio; REC, relative eosinophil count; RLC, relative lymphocyte count; sCD163, soluble cluster of differentiation 163; SNP, single nucleotide polymorphism; TNF-α, tumor necrosis factor-alpha; WBC, white blood cell.

**Table 1 cancers-15-01629-t001:** Summary of studies on the clinical use of autoantibodies as predictive biomarkers for immune-related adverse events regardless of the specific organ-autoantibody pairing.

Type of Parameter	Design (No. Patients)	Type of Tumor	Type of irAE	Main Findings	Reference
ANA, RF and ATA (detected before ICI initiation)	Retrospective (*n* = 137)	NSCLC	All types	Autoantibodies were associated with a higher risk of irAEs (OR 3.25, *p* = 0.001)	Toi Y. JAMA Oncol 2019 [27]
ANA (detected before ICI initiation)	Retrospective (*n* = 83)	NSCLC	All types	ANA were not associated with irAEs, though the risk of irAEs tended to be higher with higher titers of ANAs	Yoneshima Y. Lung Cancer 2019 [28]
ANA (detected before ICI initiation)	Retrospective (*n* = 191)	Pan-tumor	All types	ANA were not associated with irAEs, except for colitis (22% vs. 1.6%, *p* = 0.002)	Sakakida T. Clin Transl Oncol 2020 [29]
ANA, anti-dsDNA antibody, ENA *, RF, ACPA, ASMA, AMA, anti-LKM antibody and ATA (developing after ICI initiation)	Retrospective (*n* = 133)	Melanoma	All types	The association between irAEs and seroconversions was nonsignificant considering all irAEs and any autoantibody (OR 2.92, *p* = 0.12), but became significant when focusing on irAEs related to the autoantibodies tested (OR 3.64, *p* = 0.04)	de Moel EC. Cancer Immunol Res 2019 [30]
ANA, ENA and ASMA (developing after ICI initiation, within 30 days)	Retrospective (*n* = 92)	NSCLC	All types	Early detection of autoantibodies was associated with a higher risk of irAEs (HR not available, *p* = 0.002)	Giannicola R. Mol Clin Oncol 2019 [31]
ATA (titer increase from baseline)	Prospective (*n* = 78)	Pan-tumor	All types	Increases in anti-Tg and anti-TPO titers ≥ 1.5 from baseline were associated with irAE occurrence (OR 17.4, *p* = 0.015; OR 6.1, *p* = 0.035; respectively)	Music M. F1000Res 2020 [14]
ANA, RF, ATA and ANCA (before and after ICI initiation)	Retrospective (*n* = 69)	Pan-tumor	All types	Positivity for any autoantibody was associated with a higher risk of irAEs (OR 46.61, *p* = 0.010)	Les I. Ann Med 2021 [25]
ANA (detected before ICI initiation)	Retrospective (*n* = 68)	Urothelial carcinoma	All types	Patients with ANA positivity at a titer >1:160 developed irAEs more frequently (*p* = 0.029) and earlier (*p* = 0.052)	Castel-Ajgal Z. Clin Genitourin Cancer 2022 [32]
ANA, ENA **, RF, ACPA, autoimmune hepatopathy profile ^#^ and myopathy profile ^†^ (detected before ICI initiation)	Prospective (*n* = 44)	Pan-cancer	All types	The frequency of irAEs did not differ as a function of positivity for any autoantibody (OR 0.62, *p* = 0.480) or ANA titers (OR 0.79, *p* = 0.529)	Barth DA. Cancer Med 2022 [33]
ANA and ATA (detected before ICI initiation)	Retrospective (*n* = 159)	NSCLC	All types	ANA titer ≥ 1:320 was related to irAEs (OR 4.9, *p* = 0.01), especially to skin subtypes (9.7% in patients with ANA <1:320 vs. 32% in patients with ANA ≥ 1:320, *p* = 0.003)	Zhang D. Transl Lung Cancer Res 2022 [34]
ANA, anti-Ro52 and ATA (detected before ICI initiation)	Retrospective (*n* = 177)	Pan-tumor	All types	ANA and anti-Ro52 positivity was not associated with a higher risk of irAEs. ATA positivity was more common in patients with than without thyroiditis (75% vs. 13.8%, *p* < 0.001)	Tang H. Front Immunol 2022 [35]
ANA, ATA, AGAD, AChR and PA-IgG (detected before ICI initiation)	Retrospective (*n* = 275)	Pan-tumor	All types	There were no associations between autoantibodies and irAEs, except between ATA and thyroiditis (39.5% in anti-Tg-positive vs. 12.5% in anti-Tg-negative patients, *p* < 0.01)	Izawa N. ESMO 2022 [36]
ANA (detected before ICI initiation)	Retrospective (*n* = 266)	NSCLC	All types	There were no significant differences in the frequency of irAEs between positive and negative ANA patients and between high and low ANA titers	Mouri A. Front Oncol 2021 [37]
ANA (before and after ICI initiation)	Prospective (*n* = 152)	Pan-tumor	All types	There was no association between irAEs and ANA at baseline or developing. Patients who became ANA-positive during follow-up were more likely to have severe irAEs than those who were ANA-positive at baseline and ANA-negative patients (42.8% vs. 26.1% vs. 9.1%, *p* = 0.05)	Alserawan L. Int J Mol Sci 2022 [38]
ANA, RF and ACPA (before and 6 weeks after ICI initiation)	Prospective (*n* = 60)	Melanoma	All types	There was no association between baseline seropositivity for ANA/RF/ACPA and time to first irAE (*p* = 0.39). ANA/RF/ACPA-negative patients experienced more thyroid irAEs than ANA/RF/ACPA-positive patients (*p* = 0.006)	Gosh N. J Immunother Cancer 2022 [39]

* Included anti-U1RNP, anti-SS-A/Ro, anti-SS-B/La, anti-centromere protein B, anti-Scl-70, anti-Jo-1, and anti-Sm. ** Included anti-centromere protein B, anti-double strand DNA, anti-SS-B/La, anti-PM100, anti-PM75, anti-RNP70, anti-SS-A/Ro, anti-Scl-70, and anti-U1RNP. ^#^ Included anti-GP210, anti-LKM1, anti-M2, anti-SP100, anti-SLA-LP, anti-LC1, and anti-F-Actin. ^†^ Included anti-EJ, anti-Jo-1, anti-Ku, anti-MDA5, anti-MI2a, anti-MI2b, anti-NXP2, anti-Oj, anti-PL-12, anti-PL-7, anti-SAE, anti-SRP, and anti-TIF-1γ. Abbreviations in alphabetical order: ACPA, anti-citrullinated peptide antibody; AGAD, anti-glutamic acid decarboxylase antibody; AChR, anti-acetylcholinesterase receptor antibody; ALKM, anti-liver-kidney microsomal; AMA, anti-mitochondrial antibody; ANA, antinuclear antibody; ANCA, antineutrophil cytoplasmic antibody; ASMA, anti-smooth muscle antibody; ATA, antithyroid antibody; dsDNA, double-stranded DNA antibody; ENA, extractable nuclear antigen antibody; HR, hazard ratio; ICI, immune-checkpoint inhibitor; irAE, immune-related adverse event; NSCLC, non-small lung cancer; OR, odds ratio; PA-Ig, platelet-associated immunoglobulin G; RF, rheumatoid factor; Tg, thyroglobulin; TPO, thyroid peroxidase.

**Table 2 cancers-15-01629-t002:** Summary of studies on the clinical use of blood cell counts and ratios as predictive biomarkers for immune-related adverse events.

At Baseline (Before Immune-Checkpoint Inhibitor Initiation)					
Type of Parameter	Study Design (No. Patients)	Type of Tumor	Type of irAE	Main Findings	Reference
ALC	Retrospective (*n* = 167)	Pan-tumor	All types	Grade ≥ 2 irAEs were associated with ALC > 2000/μL (OR 1.996, *p* < 0.05)	Diehl A. Oncotarget 2017 [41]
AEC	Retrospective (*n* = 45)	Melanoma	Endocrine irAEs	irAEs were associated with AEC > 240/μL (OR 1.601, *p* = 0.045)	Nakamura Y. Jpn J Clin Oncol 2019 [42]
AEC	Retrospective (*n* = 95)	Pan-tumor	All types	AEC > 0.045 × 10^9^/L was predictive of irAEs (OR 4.114, *p* = 0.014)	Ma Y. World J Surg Oncol 2022 [43]
NLR	Prospective (*n* = 1187)	Pan-tumor (blood and solid organ cancers)	All types	NLR > 4.78 was predictive of grade 4 and 5 irAEs (OR not available, *p* = 0.0137) *	Ruste V. Eur J Cancer 2021 [44]
NLR and PLR	Retrospective (*n* = 184)	NSCLC	All types	PLR < 180 was the only independent predictor of irAEs (OR 2.3, *p* = 0.017)	Pavan A. Oncologist 2019 [45]
dNLR	Retrospective (*n* = 391)	Pan-tumor	All types	dNLR ≥ 3 was protective against irAEs (OR 0.37, *p* = 0.012)	Eun Y. Sci Rep 2019 [46]
ANC, PC, NLR and PLR	Retrospective (*n* = 150)	NSCLC	All types	Grade 3–4 irAEs were associated with ANC (*p* = 0.009), PC (*p* = 0.023), NLR (*p*= 0.023) and PLR (*p* = 0.0016) * (cut-off values and ORs not provided)	Liu W. Cancer Manag Res 2021 [47]
RLC and AEC	Retrospective (*n* = 105)	Pan-tumor	All types	irAEs were associated with RLC < 28.5% (OR 3.60, *p* = 0.027) and AEC > 0.175 × 10^9^/L (OR 3.44, *p* = 0.020)	Bai R. Cancer Biol Med 2021 [48]
NLR	Retrospective (*n* = 115)	NSCLC	All types	irAEs were associated with NLR < 2.86 (OR 2.69, *p* = 0.016)	Fujimoto A. Thorac Cancer 2021 [49]
ALC, AMC, APC, NLR, MLR and PLR	Retrospective (*n* = 470)	Pan-tumor	All types	irAEs were associated with ALC > 2.6 K/μL (aOR 4.3, *p* = 0.002), AMC > 0.29 K/μL (aOR 2.34, *p* = 0.03), PC > 145 K/μL (aOR 2.23, *p* = 0.03), NLR ≤ 5.3 (aOR 2.07, *p* = 0.01), MLR ≤ 0.76 (aOR 2.96, *p* = 0.01) and PLR ≤ 534 (aOR 5.05, *p* = 0.04) **	Michailidou D. Sci Rep 2021 [50]
NLR	Metanalysis ^#^ (*n* = 6696)	NSCLC	All types	irAEs were associated with NLR ≥ 5 (OR = 1.046, *p* = 0.026)	Suazo-Zepeda E. Cancer Immunol Immunother 2021 [51]
ALC, LMR, NLR and PLR	Retrospective (*n* = 92)	NSCLC	All types	ALC > 1450/mm^3^ (aOR 0.24, *p* = 0.003) and LMR > 1.6 (OR 0.12, *p* = 0.004) were associated with a lower risk of irAEs. NLR > 2.3 (aOR 5.99, *p* = 0.005) and PLR > 165 (OR = 2.87, *p* = 0.022) were associated with a higher risk of irAEs ^†^	Egami S. J Cancer 2021 [52]
ALC	Retrospective (*n* = 667)	NSCLC	All types	ALC was positively associated with irAE risk (OR 2.556, *p* = 0.001; ALC cut-off value not provided)	Xu H. Exp Cell Res 2022 [53]
NLR	Retrospective (*n* = 147)	Pan-tumor	All types	NLR < 3 was associated with a higher rate of irAE (aOR 2.27, *p* = 0.034)	Lee PY. Cancers (Basel) 2021 [54]
AEC	Retrospective (*n* = 300)	NSCLC	Pneumonitis	Pneumonitis was associated with AEC ≥ 0.125 × 10^9^/L (HR 2.825, *p* < 0.001)	Chu X. Lung Cancer 2020 [55]
ALC	Retrospective (*n* = 110)	Pan-tumor	Myocarditis	ALC 1.6 K/μL in myocarditis group vs. 1.3 K/μL in non-myocarditis group (*p* = 0.02) *	Drobni ZD. J Am Heart Assoc 2020 [56]
NLR	Retrospective (*n* = 73)	Gastric and renal cancers	Grade 3 and 4 irAEs	NLR < 4.3 was associated with lower risk of grade 3–4 irAEs (OR 0.024, *p* = 0.014)	Takada S. Asian Pac J Cancer Prev 2022 [57]
During Follow-Up (After Immune-Checkpoint Inhibitor Initiation)					
Type of Parameter	Study Design	Type of Tumor	Type of irAE	Main Findings	Reference
WBC RLC (on the day of irAE detection)	Retrospective (*n* =101)	Melanoma	Lung and gastrointestinal irAEs	59.1% increase in WBC (OR = 6.04, *p* = 0.014) and 32.3% decrease in RLC (OR = 5.01, *p* = 0.012) were predictive of irAEs	Fujisawa Y. J Dermatol Sci 2017 [58]
ALC at 1 month	Retrospective (*n* = 167)	Pan-tumor	All types	Grade ≥ 2 irAEs were associated with ALC > 2000/μL (OR = 1.813, *p* < 0.05)	Diehl A. Oncotarget 2017 [41]
REC at 1 month WBC at 1 month	Retrospective (*n* = 45)	Melanoma	Endocrine irAEs and vitiligo	REC > 3.2% was predictive of irAEs (OR = 5.111, *p* = 0.025) *. A summative increase in WBC by 100 was protective against vitiligo (OR = 0.823, *p* = 0.0023).	Nakamura Y. Jpn J Clin Oncol 2019 [42]
ANC NLR PLR (treatment cycle before onset of the irAE)	Retrospective (*n* = 150)	NSCLC	All types	Multiple univariate associations were described, namely, between *: 1.91 × 10^9^/L decrease in ANC and grade 1–2 irAEs (*p* = 0.013) 1.11 × 10^9^/L decrease in ANC and grade 3–4 irAEs (*p* = 0.003) 0.62 decrease in NLR and grade 1–2 irAEs (*p* = 0.013) 0.76 decrease in NLR and grade 3–4 irAEs (*p* = 0.011) 89.26 decrease in PLR and grade 1–2 irAEs (*p* = 0.011) (comparative data between baseline and pre-irAE cycle)	Liu W. Cancer Manag Res 2021 [47]
ALC at 2 weeks	Retrospective (*n* = 171)	NSCLC	All types	Early onset of irAEs was associated with ALC > 820/mm^3^ (aOR = 3.58, *p* = 0.07) ^†^	Egami S. Front Oncol 2021 [59]
NLR at second course (2 to 3 weeks after the first dose)	Retrospective (*n* = 243)	Esophageal, gastric and colon cancer	All types	irAEs (any grade) were associated with NLR < 3 (OR = 0.894, *p*= 0.044)	Zhang Z. Cancers (Basel) 2022 [60]
ALC NLR (from baseline to last ICI dose; and from baseline to myocarditis onset)	Retrospective (*n* = 110)	Pan-tumor	Myocarditis	irAEs were associated with a decrease in ALC (1.6 K/μL to 1.4 K/μL to 1.1 K/μL, *p* < 0.001) and an increase in NLR (3.5 to 4.1 to 6.6, *p* < 0.001) *	Drobni ZD. J Am Heart Assoc 2020 [56]
NLR (at the onset of the irAE)	Retrospective (*n* = 73)	Gastric and renal cancers	Grade 3 and 4 irAEs	∆NLR >120% was associated with increased risk of irAEs (OR = 10.48, *p* = 0.033)	Takada S. Asian Pac J Cancer Prev 2022 [57]

* Only in the univariate analysis. ** All odds ratios adjusted for age, sex, smoking history, cancer type (hematological malignancy vs. solid tumor), Eastern Cooperative Oncology Group performance status, concomitant systemic therapy, personal or family history of autoimmune disease, personal history of chronic infection, systemic steroid treatment at the time of ICI initiation. ^#^ Including randomized controlled trials, cohort and case-control studies. Abbreviations in alphabetical order: AEC: absolute eosinophil count; ALC: absolute lymphocyte count; AMC: absolute monocyte count; ANC: absolute neutrophil count; APC: absolute platelet count; dNLR (calculated as absolute neutrophil count/[white blood cell count–absolute neutrophil count]): derived neutrophil-to-lymphocyte ratio; LMR: lymphocyte-to-monocyte ratio; MLR: monocyte-to-lymphocyte ratio; NLR: neutrophil-to-lymphocyte ratio; NSCLC, non-small cell lung cancer; PC: platelet count; PLR: platelet-to-lymphocyte ratio; REC: relative eosinophil count; RLC: relative lymphocyte count; WBC: white blood cell. ^†^ All odds ratios adjusted for age and PD-L1 expression (≥50%). ∆: rate of change.

**Table 4 cancers-15-01629-t004:** Summary of gut microbiome members related to the risk of developing immune-related adverse events.

Type of Taxonomic Category or Microorganism	Type of Effect	Type of irAE Assessed	Related References
*Bacteroidetes* phylum *	Protective factor	Colitis	Dubin K. Nat Commun 2016 [191] Chaput N. Ann Oncol 2019 [63] Liu T. Immunotherapy 2019 [194] Sakai K. Front Oncol 2021 [186] Liu W. Front Immunol 2021 [195]
*Bacteroidetes* phylum	Protective factor	Pancreatic irAEs	Tan B. Thorac Cancer 2021 [196]
*Bacteroides dorei* *Bacteroides vulgatus*	Risk factor Protective factor	General irAEs	Usyk M. Genome Med 2021 [197]
*Bacteroides fragilis* *Bacteroides thetaiotaomicron*	Protective factor	Colitis (in mice)	Vétizou M. Science 2015 [198]
*Bacteroides thetaiotaomicron* *Bacteroides faecis*	Risk factor	Myocarditis	Gil-Cruz C. Science 2019 [199]
*Bacteroides intestinalis*	Risk factor	General irAEs (grade ≥ 3)	Andrews MC. Nat Med 2021 [109]
*Prevotellamassilia timonensis* (from *Bacteroidetes* phylum)	Risk factor	Severe colitis	Mao J. J Immunother Cancer 2021 [190]
*Firmicutes* phylum **	Risk factor	Colitis	Dubin K. Nat Commun 2016 [191] Chaput N. Ann Oncol 2019 [63] Gopalakrishnan V. Science 2018 [188] Liu T. Immunotherapy 2019 [194]
*Firmicutes* phylum	Risk factor	Pancreatic irAEs	Tan B. Thorac Cancer 2021 [196]
*Phascolarctobacterium* genus (from *Firmicutes* phylum)	Protective factor	Colitis	Liu T. Immunotherapy 2019 [194]
*Faecalibacterium* genus (from *Firmicutes* phylum)	Protective factor	Absent or grade 0–2 colitis	Liu W. Front Immunol 2021 [195]
*Bifidobacterium* *Bifidobacterium breve ^#^*	Protective factor	Colitis (in mice)	Wang F. Proc Natl Acad Sci USA 2018 [200] Sun S. Proc Natl Acad Sci USA 2020 [201]
*Bifidobacterium*	Protective factor	General irAEs	Chau J. J Clin Oncol 2021 [193]
*Lactobacillus rhamnosum*	Protective factor	Colitis (in mice)	Sun S. Proc Natl Acad Sci USA 2020 [201]
*Lactobacillaceae* family *Raoultella* genus *Akkermansia* species *Agathobacter* genus	Protective factor Protective factor Protective factor Risk factor	Low-grade irAEs Low-grade irAEs Low-grade irAEs High-grade irAEs	Hakozaki T. Cancer Immunol Res 2020 [202]
*Enterobacteriaceae* family ^†^	Protective factor (remission of colitis)	Colitis	Sakurai T. Mol Oncol 2022 [189]
*Intestinibacter barlettii* *Anaerotignum lactatifermentans* *Dorea formicigenerans*	Risk factor Protective factor Protective factor	General irAEs (grade ≥ 3)	Andrews MC. Nat Med 2021 [109]
*Streptococcus* genus *Paecalibacterium* genus *Stenotrophomonas* genus	Risk factor	General irAEs (grade ≥ 3)	Liu W. Front Immunol 2021 [195]
*Lachnospiraceae* species *Streptococcaceae* species	Risk factor	General irAEs	McCulloch JA. Nat Med 2022 [192]
*Akkermansia muciniphila*	Protective factor	Colitis	Wang L. Gut 2020 [203]
*Alispides* genus	Protective factor	Pancreatic irAEs	Tan B. Thorac Cancer 2021 [196]
*Lachnospiraceae* genus	Risk factor	Pancreatic irAEs	Tan B. Thorac Cancer 2021 [196]
*Burkholderia* cepacia	Protective factor	Colitis in mice	Vétizou M. Science 2015 [198]
*Proteobacteria* phylum			
*Desulfovibrio*	Protective factor	General irAEs	Chau J. J Clin Oncol 2021 [193]
*Veillonela*	Risk factor	Colitis	Liu T. Immunotherapy 2019 [194]
*Staphylococcus epidermidis*	Risk factor	Dermatitis (in mice)	Hu ZI. Proc Natl Acad Sci USA 2022 [204]

* Including Bacteroidaceae, Rikenellaceae and Barnesiellaceae families. ** Including Ruminococcaceae, Clostridium cluster XIVa, Blautia, and Faecalibacterium. ^#^ Only for Bifidobacterium breve, but not for other Bifidobacterium strains. ^†^ Including operational taxonomic units classified as Shigella flexneri, Citrobacter, Klebsiella pneumoniae, Enterobacter cloacae, and other unclassified Enterobacteriaceae. Abbreviations in alphabetical order: irAE, immune-related adverse event.

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
