# Peer review of "Predictive Biomarkers for Checkpoint Inhibitor Immune-Related Adverse Events"

_cancers, 2023, doi:10.3390/cancers15051629_

Round 1

Reviewer 1 Report

First - it should be noted that this is a review article, therefore not graded higher on the above 'originality or contribution to the field' section. Having said that, I think the article is an excellent summary of the predictive biomarkers of IrAEs and has been presented in an detailed, yet easy-to-read manner and touches upon all the important points in the field and is up to date on latest literature. The illustrations are also well presented. My only recommendation would be to reduce the length of the article, if at all possible, I think it is a little too long in its current state. 

Author Response

Thank you very much for your comments.

We have reduced the length of the article by moving the "Materials and Methods" section to supplementary material.

Reviewer 2 Report

In the present review authors reviewed the biomarker that will predict the immune related adverse even in cancer patients treated with immune checkpoint blockade therapy. Author’s reviewed number of papers for biomarkers predicting immune related adverse events such as autoantibodies, blood cell count and ratios, cytokines, HLA genotyping, micro-RNAs, Gut microbiome. however, the manuscript is superficial and lacks deep insight. Reviews are not mere summary of the literature but criticizes important concepts, ideas, issues, and problems. Therefore, the Reviewer believes that the manuscript is not sufficient for the publication in the high-quality journals such as Cancers.

1.      Reviewer think that  (Section 2) Materials, methods and (section 3) results as well as figure 1 are not necessary for this review article, readers will be distracted and not engaged to the main idea of the review.

2.     Criticize about line 183-285 why this is happening

3.     Line 222. It is necessary to boost immune system after ICB thus CD8 expansion is necessary to mount anti-tumor immune response. Why author think that CD8 t cells expansion and diversification can predict irAE.

4.     Similar to above question how to differentiate between the level of C-reactive protein of irAE and autoimmune disease, as there are number of patients who have both autoimmune disease and cancer and C-reactive protein in predictive marker to arthritis.

5.     Line 369-372 highlight more insight about why single polymorphism in PDCD1 gene is associated with both protective and susceptibility biomarkers.

6.     Line 449, different miRs performs different function at the same some of them are pleotropic in nature. Please elaborated what is authors thought behind highlighting only downregulation of innate and adaptive immune response.

7.     Please work on figure 2 resolution, colors, fonts. 

Author Response

  1. Reviewer think that  (Section 2) Materials, methods and (section 3) results as well as figure 1 are not necessary for this review article, readers will be distracted and not engaged to the main idea of the review.

Response: We have reduced the length of the article by removing the "Materials and Methods" section and placing it in the supplementary materials.

  1. Criticize about line 183-285 why this is happening.

Response: Could the reviewer specify the sentence(s) (not the number line) that he/she would like to be clarified or criticized please?

  1. Line 222. It is necessary to boost immune system after ICB thus CD8 expansion is necessary to mount anti-tumor immune response. Why author think that CD8 t cells expansion and diversification can predict irAE.

Response: Increased T-cell clonality in the tumor microenvironment is a previously reported factor associated with clinical response to anti-PD-1 therapy. Since tumor response induced by immune checkpoint inhibitors (ICIs) depends on the action of immune effector cells, mainly CD8 T-cells, it may be that T-cell responses in the tumor sites would be beneficial. However, an increase in the T-cell clonality in the systemic circulation may also indicate a response against self-antigens, which may lead to toxicities. Indeed, CD8 T-cell clonal expansion has been demonstrated to precede the development of irAEs.

Likewise, increased CD8 T-cell diversity would support a mechanism of auto-reactivity against multiple self-antigens induced by ICI administration, by mobilization of different T-cell clones, which may also lead to irAE development.

To facilitate the understanding of this part of the manuscript, we have changed the sentence as follows: “Similarly, it has been reported that peripheral CD8 T-cell expansion and diversification after ipilimumab initiation, as surrogate markers of autoreactivity against tissue self-antigens at the systemic level, allows us to predict irAE onset with very high sensitivity [62]”.

  1. Similar to above question how to differentiate between the level of C-reactive protein of irAE and autoimmune disease, as there are number of patients who have both autoimmune disease and cancer and C-reactive protein in predictive marker to arthritis.

Response: It is not possible to distinguish whether the CRP elevation is due to an immune-related adverse event (irAE) or to a reactivation of a previous autoimmune or inflammatory disease, such as arthritis.

For this reason, we have changed the general definition given for an irAE in the first paragraph of the “Introduction”:

“In practical terms, an irAE can be defined as any symptom, sign, syndrome, or disease caused or exacerbated by an immune-activating mechanism during the administration of an ICI once other causes such as infectious diseases or tumor progression have been ruled out”.

  1. Line 369-372 highlight more insight about why single polymorphism in PDCD1 gene is associated with both protective and susceptibility biomarkers.

Response: Bins S et al. showed that patients harboring two variant alleles in the rs2227981 SNP of the PDCD1 gene have shown a lower susceptibility for irAEs. An explanation for this seemingly paradoxical finding could be that these patients with two variant alleles have less PD-1-mediated T cell inhibition, which may make this pathway less susceptible to blockade by PD-1 inhibitors such as nivolumab. Consequently, these patients may experience fewer treatment-related toxicities than wildtype patients. In contrast, Kobayashi M et al. found that rs10204525 SNP of the same gene (PDCD1) was related to an increase in PD-1 expression. In general, a high PD-1 expression seems to be related to a substantial antitumor effect and the development of irAEs induced by ICI administration.

To facilitate the understanding of this part of the manuscript, we have rewritten the sentence as follows: “In the same vein, two different PDCD1 gene single-nucleotide polymorphisms (SNPs), namely rs2227981 and rs10204525, have been identified as both protective and susceptibility biomarkers for irAEs, respectively; these apparently paradoxical findings are explained by the polymorphism in question, which determines the level of PD-1 expression (low in the case of rs2227981 and high in the case of rs10204525).”

  1. Line 449, different miRs performs different function at the same some of them are pleotropic in nature. Please elaborated what is authors thought behind highlighting only downregulation of innate and adaptive immune response.

Response: We thank the reviewer for this comment. With this sentence, we refer to the role of certain miRs, such as miR-146a, which is crucial for regulatory T (Treg) cell functions. Indeed, deficiency of miR-146a resulted in massive lymphocyte activation, tissue infiltration and immune-mediated damage. Therefore, we have rewritten the sentence as follows: “Like CTLA-4 and PD-1, certain miRs, such as miR-146a, promote down-regulation of both innate and adaptive immune responses and may impact ICI-related survival [172], in part by counteracting cell escape mechanisms in the tumor microenvironment [173]”.

  1. Please work on figure 2 resolution, colors, fonts.

Response: We have submitted a new improved version of Figure 2 attached to the manuscript.

Reviewer 3 Report

In this manuscript, the authors reviewed the literature on predictive biomarkers for immune-related adverse events (irAEs) developing under Immune-checkpoint inhibitors (ICIs), differentiating between biomarkers already available for daily practice and those still at the research stage. As biomarkers they discuss the available data on Autoantibodies, Blood cell counts and ratios, Serum and other biological fluid proteins, Cytokines, and Biomarkers under investigation. They concluded that it is difficult to establish the application of irAE biomarkers based on the current evidence because most studies have been retrospective, time-limited and restricted to a specific type of cancer, irAE or ICI. Long-term prospective cohorts and real-life studies are therefore needed to assess the predictive capacity of different potential irAE biomarkers, regardless of the ICI type, organ involved, or cancer site. The study is of current interest and improvement in AEs development and management are needed to improve the clinical benefit of the increasing ICIs in oncology.  As stated by the authors irAEs may mimic traditional autoimmune disorders. However, the authors should further stress an important difference between classical autoimmune disease and irAEs discussing the diagnostic role of autoantibodies. When they discuss the role of autoantibodies, in particular the role of ANA, they should the diagnostic relevance of the different ANA immunofluorescence patterns. For instance, it is well-known that ICIs may cause liver toxicity potentially mimicking autoimmune liver disease (autoimmune hepatitis, primary biliary cholangitis). However, they should recall that classic autoimmune hepatitis is characterized by more disease-specific ANA pattern such as ANA with a "homogenous" immunofluorescence pattern, while primary biliary cholangitis are associated with the so-called ANA PBC-specific ("multiple nuclear dots" and "rim-like/membranous" as previously described (Diagnosis and therapy of autoimmune hepatitis. Mini Rev Med Chem. 2009 Jun;9(7):847-60; Antinuclear antibodies giving the 'multiple nuclear dots' or the 'rim-like/membranous' patterns: diagnostic accuracy for primary biliary cirrhosis. Aliment Pharmacol Ther. 2006 Dec;24(11-12):1575-83; Autoantibodies to speckled protein family in primary biliary cholangitis. Allergy Asthma Clin Immunol. 2021 Mar 31;17(1):35; ).

-Another important topic is the pathogenic role of Regulatory Foxp3+ T cell a subset of t cells with a crucial role in avoiding autoimmunity but also the most abundant immunosuppressive role in the tumor microenvironment of different cancers such as hepatocellular carcinoma. Since they express PD1 and CTLA-4 thus being a potential target of ICIs, it has been recently suggested a role of this immune cell subset in the development of irAEs as recently described (Hepatocellular carcinoma in viral and autoimmune liver diseases: Role of CD4+ CD25+ Foxp3+ regulatory T cells in the immune microenvironment. World J Gastroenterol. 2021 Jun 14;27(22):2994-3009.).

Author Response

In this manuscript, the authors reviewed the literature on predictive biomarkers for immune-related adverse events (irAEs) developing under Immune-checkpoint inhibitors (ICIs), differentiating between biomarkers already available for daily practice and those still at the research stage. As biomarkers they discuss the available data on Autoantibodies, Blood cell counts and ratios, Serum and other biological fluid proteins, Cytokines, and Biomarkers under investigation. They concluded that it is difficult to establish the application of irAE biomarkers based on the current evidence because most studies have been retrospective, time-limited and restricted to a specific type of cancer, irAE or ICI. Long-term prospective cohorts and real-life studies are therefore needed to assess the predictive capacity of different potential irAE biomarkers, regardless of the ICI type, organ involved, or cancer site. The study is of current interest and improvement in AEs development and management are needed to improve the clinical benefit of the increasing ICIs in oncology.  As stated by the authors irAEs may mimic traditional autoimmune disorders. However, the authors should further stress an important difference between classical autoimmune disease and irAEs discussing the diagnostic role of autoantibodies. When they discuss the role of autoantibodies, in particular the role of ANA, they should the diagnostic relevance of the different ANA immunofluorescence patterns. For instance, it is well-known that ICIs may cause liver toxicity potentially mimicking autoimmune liver disease (autoimmune hepatitis, primary biliary cholangitis). However, they should recall that classic autoimmune hepatitis is characterized by more disease-specific ANA pattern such as ANA with a "homogenous" immunofluorescence pattern, while primary biliary cholangitis are associated with the so-called ANA PBC-specific ("multiple nuclear dots" and "rim-like/membranous" as previously described (Diagnosis and therapy of autoimmune hepatitis. Mini Rev Med Chem. 2009 Jun;9(7):847-60; Antinuclear antibodies giving the 'multiple nuclear dots' or the 'rim-like/membranous' patterns: diagnostic accuracy for primary biliary cirrhosis. Aliment Pharmacol Ther. 2006 Dec;24(11-12):1575-83; Autoantibodies to speckled protein family in primary biliary cholangitis. Allergy Asthma Clin Immunol. 2021 Mar 31;17(1):35; ).
Response: We agree with the reviewer's comment on the differences in the ANA pattern between classic autoimmune diseases and immune-mediated adverse events. In this regard, we have added the following sentence in the text: “In addition to ANA positivity, ANA patterns determined by immunofluorescence may be useful for discriminating between primary autoimmune diseases and irAEs. Although no studies designed to compare autoantibody profiles have yet been reported, patients who develop irAEs may be less likely to express disease-specific ANA patterns than patients with the corresponding classical autoimmune disease. In contrast, a nuclear speckled pattern may be more indicative of immune-related toxicity [38]”.

We have also added a new reference supporting this statement.

-Another important topic is the pathogenic role of Regulatory Foxp3+ T cell a subset of t cells with a crucial role in avoiding autoimmunity but also the most abundant immunosuppressive role in the tumor microenvironment of different cancers such as hepatocellular carcinoma. Since they express PD1 and CTLA-4 thus being a potential target of ICIs, it has been recently suggested a role of this immune cell subset in the development of irAEs as recently described (Hepatocellular carcinoma in viral and autoimmune liver diseases: Role of CD4+ CD25+ Foxp3+ regulatory T cells in the immune microenvironment. World J Gastroenterol. 2021 Jun 14;27(22):2994-3009.).

Response: We agree with the reviewer’s comment on the pathogenic role of regulatory Foxp3+ T cells in the emergence of immune-related toxicity. However, the current review article aims to collect those biomarkers that have been investigated with the intention of predicting irAEs. Since this cellular subtype could be an interesting line of research in the future, the following sentence is included at the end of section "Blood cell counts and ratios": “Likewise, it has been suggested that changes in the percentage of peripheral CD4+ CD25+ Foxp3+ regulatory T cells, a cell subset in charge of maintaining immune tolerance in the tumor microenvironment, may be predictive of irAEs [62]”. 

We have also added the reference proposed by the reviewer in the manuscript.

Round 2

Reviewer 2 Report

  1. Criticize about line 183-185 why this is happening. The reviewer's would like the authors to criticize about why  there is differential irAE response to preform antibody and seroconverted ANA autoantibody 

Author Response

As indicated in the text, patients experiencing seroconversion to ANA positivity after ICI initiation were more predisposed to high-grade irAEs than patients who remained ANA-negative and patients who were ANA-positive before ICI initiation. Regarding this last subgroup, the pre-existence of ANA before ICI initiation may indicate an autoimmune predisposition that does not necessarily imply the development of an autoimmune complication. Indeed, a study by Gosh et al. showed that patients with lower baseline autoantibody levels including ANAs are at a higher risk of developing irAEs during follow-up. On the contrary, de novo development of ANAs under treatment with ICIs could be related to early B-cell changes, specifically a decline in circulating B cells and an increase in CD21 B cells and plasmablasts, which have been associated with a higher frequency of irAEs.

We have added a new sentence explaining this finding with a new reference: “This humoral response in the form of ANA seroconversion could be related to early B-cell changes induced by ICIs, namely a decline in circulating B cells and an increase in CD21 B cells and plasmablasts, which have been associated with a higher frequency of irAEs (40)”

Reviewer 3 Report

The authors addressed the raised points.

Author Response

Thank you very much for your comments.